# FACTORED REPRESENTATION FOR NEURO-SYMBOLIC AI

## ABSTRACT

Large language models (LLMs) have demonstrated remarkable capabilities across a wide range of tasks, yet their internal decision-making processes remain largely opaque, posing a significant challenge to their trustworthiness in complex reasoning scenarios. We explore the hypothesis that compelling a pre-trained LLM to maintain a structured internal state using a formal, symbolic representation can enhance interpretability without degrading its reasoning performance. To investigate this, we employ two primary techniques: prompt fine-tuning and parameter-efficient fine-tuning (PEFT) using LoRA. Thus, we prompt a variety of LLMs to articulate their reasoning steps using various structured formalisms, including basic semantic triples, lists of attribute-value pairs, and first-order logic. We also fine-tune a pre-trained LLM on a structured representation that the LLM is subsequently prompted to use as an internal representation during reasoning to solve a task. Our results demonstrate that while state-of-the-art models struggle to generate consistently structured reasoning, their core reasoning capabilities remain largely intact. This suggests that the LLM's reasoning mechanism is not necessarily fully aligned with its generative capabilities. However, this result shows the potential for specialized models capable of performing complex reasoning while providing verifiable chains of thought.

## 1 INTRODUCTION

Large language models (LLMs) have demonstrated remarkable capabilities in a wide range of tasks, yet their internal decision-making processes remain largely opaque and unverifiable (Mohebbi et al., 2024). This lack of clarity poses a significant challenge to their trustworthiness, particularly in complex reasoning tasks where reliability and interpretability are paramount. When a model fails, it is difficult to diagnose the error, and when it succeeds, it is hard to verify that its success stems from genuine comprehension rather than spurious correlations.

The field of Neuro-Symbolic AI (Garcez & Lamb, 2023; Bader & Hitzler, 2005) seeks to bridge the gap between data-driven neural networks and symbolic systems. This approach aims to create more robust and capable models by combining the fast, perceptual abilities of neural networks with the structured, explicit logic of symbolic reasoning. A key focus within this domain is knowledge representation — how facts, concepts, and the relationships between them are formalized to connect natural language to semantic meaning. By integrating symbolic logic and compelling LLMs to externalize their reasoning process into a formal structure, we can make models' "thought processes" more transparent.

This project explores the hypothesis that forcing a pre-trained LLM to maintain a structured internal state using a formal representation can lead to comparable reasoning performance, but with significantly improved interpretability. Investigate whether explicitly articulating reasoning steps in a symbolic format can make a model's thought process more transparent and verifiable. We experiment with several formalisms, including semantic triples, First-Order Logic, Propositonal Logic, and alists (Associative Logic for Inference, Storage and Transfer) (Nuamah & Bundy, 2023) a recursive, expressive formalism designed for querying across diverse knowledge bases.

To test our hypothesis, we evaluate the performance of the Qwen3-4B, Qwen3-8B, and the gpt-oss-20b models (Qwen, 2025; OpenAI, 2025d) on key reasoning benchmarks. Our methodology involves two main strategies: prompt fine-tuning, where we guide models to utilize these formalisms

at each step of its thinking via zero-shot, one-shot, and few-shot prompts , and parameter-efficient fine-tuning (PEFT), where we use LoRA (Hu et al., 2021) to fine-tune the Qwen3-4B model on an LLM-generated dataset of natural language and alist pairs.

Our findings demonstrate that constraining models to a formal representation has a negligible effect on their core reasoning capabilities, and show that the models largely struggle to adhere to the provided structure. Despite experimenting with various formalisms and fine-tuning the models to generate structured outputs, our results show a clear disconnect between adherence to a formal representation and reasoning accuracy. This suggests that the internal reasoning processes remain independent of these external constraints, highlighting the challenge of achieving neuro-symbolic integration in pre-trained LLMs, and the need for models highly capable in both performing complex reasoning tasks, and outputting verifiable chains of thought that are correctly aligned with their internal reasoning mechanism.

## 2 BACKGROUND

We first provide a broader overview of the field of Neuro-symboblic Artificial Intelligence (AI), and it's relation to knowledge representation and interpretability in large language models (LLMs). We then provide an explanation of our key factored representation, alists.

### 2.1 NEURO-SYMBOLIC MODELS

Neuro-Symbolic AI aims to bridge the gap between neural approaches and ruled-based reasoning by integrating symbolic reasoning into neural networks. This approach seeks to build a more robust and capable model that benefits from both the fast, data-driven perception of neural networks and the structured logic of symbolic systems, which allow for explicit use of expert knowledge, and provide interpretable results. Such approaches are increasingly used for complex reasoning tasks, such as knowledge-based question answering (Kapanipathi et al., 2021), visual question answering (Yi et al., 2019), and algorithmic question answering (Veličković & Blundell, 2021; Nuamah, 2021).

**Knowledge Representation.** A key area of research within the field of Neuro-Symbolic AI focuses on the representation of facts, questions, and phrases. More specifically, research looks at representing complex relationships between entities and improving the connection between natural language and semantic meaning in AI systems. Dynamic knowledge graphs have been shown to improve performance on multi-hop reasoning (Chen et al., 2023) and zero-shot common-sense answering (Bosselut et al., 2020). Beyond using traditional representations such as first-order logic (Badreddine et al., 2022; Riegel et al., 2020) alongside neural approaches, new representations have been used in order to better capture long-range relationships (Papoulias, 2023). Training with these representations has also shown to be more efficient than traditional methods and therefore reduce environmental impact (Colelough & Regli, 2025).

**Interpretability and Trustworthiness.** State-of-the-art models' reasoning is opaque (Mohebbi et al., 2024) and unverifiable. The lack of clarity on their decision-making processes, and their levels of comprehension when breaking down and solving complex reasoning tasks makes them unreliable on unseen, harder problems, without further verification. Neuro-symbolic methods seek to improve models' interpretability in addition to their trustworthiness when solving more difficult, domain-specific tasks, such as grading answers (Künnecke et al., 2024) or abstractive conversation summarization (Ribeiro et al., 2022).

### 2.2 LIGHTWEIGHT FORMALISMS

Knowledge graphs (KGs) have long been used both within AI systems, and in other areas of computer science, most significantly in support of search engines (Noy et al., 2019). However, despite their widespread adoption, there is often a lack of uniformity in the knowledge representation formalisms in use. Lightweight representations such as JSON-LD (Sporny et al., 2014) and *alists* (Associative Logic for Inference, Storage and Transfer) (Nuamah & Bundy, 2023) aim to provide a formalism which is expressive enough to allow for querying across knowledge bases with differing representations (e.g. SPARQL (World Wide Web Consortium's SPARQL Working Group, 2013),

first-order logic expressions). They are also meant to be easy for humans to read and write. For this work we use the alist representation for it's simplicity in the use of surface text instead of URIs to represent concepts.

An alist is recursively defined as a set of attribute-value pairs used to represent a fact or a question. Its formal definition is a set of pairs $(x, y)$ where $x \in \{\mathcal{F} \cup \mathcal{O} \cup \mathcal{M} \cup \text{variable}\}$ and $y \in \{C \cup v \cup \mathcal{A}\}$. Here, $\mathcal{F}, \mathcal{O}, \mathcal{M}$ are sets of functional, object-level, and meta-level attributes; $C$ are constants, $v$ are variables, and $\mathcal{A}$ is itself an alist. The recursive aspect of alists allows for nested alists which can represent more complex queries and aid in multi-hop reasoning.

We illustrate some examples of alists below:

*Example 1: What was the capital of Japan in 1960?*

```
{h:value, v:[?x], s:Japan, p:capital,    o:?x, t:1960}
```

where s is a subject, p is a predicate, o is an object, t is time, ?x is a variable and h is the operation to perform and v is a list of variables and arguments for h.

*Example 2: What is the country with the highest GDP in Europe?*

```
{h:argmax, v:[?x,?y], s:?, p:GDP, o:?y, l:Europe}
```

*Example 3: The comparison-based sorting algorithm with an average-case time complexity of $O(n \log n)$ and a worst-case space complexity of $O(\log n)$ is merge sort.*

```
    {h: AND,
     v: [fact_1,
        fact_2,
        fact_3,
        fact_4],
    fact_1: {o: merge sort,
            p:type,
            s: sorting algorithm},
    fact_2: {o: merge sort,
            p: type,
            s: comparison based},
    fact_3: {h: average-case,
            p: time complexity,
            s: merge sort,
            o: O(n log n)},
    fact_4: {h: worst-case,
            p: space complexity,
            s: merge sort,
            o: O(log n)}
    }
```

## 3  FACTORED REPRESENTATION IN REASONING

We explore whether forcing a pre-trained LLM to maintain a structured internal state that expresses concepts and unknowns using a formal representation leads to comparable or better reasoning performance but with improved interpretability, hence improving trustworthiness in complex reasoning tasks. By *factored representation*, we mean a structured formalism where distinct features in the environment (text or reasoning problem) are broken down into a set of variables or attributes and their values. We experiment with simple data triples, alists and (first-order) formal logic in increasing order of expressiveness and semantic complexity. We describe each of these below.

**Semantic Triples.**  Data triples are a common form of knowledge representation (Berners-Lee et al., 2001; Bizer et al., 2009). More specifically, we use ⟨subject, predicate, object⟩ triples, (Mcbride, 2004), in order to simulate a dynamic knowledge graph based on the context of the given problem.

*Semantic Triple Example:*

Ada Lovelace was born in London.

- ⟨Ada Lovelace, born in , London⟩

London is a city in England.

- ⟨London, is , English city⟩

Therefore, Ada Lovelace was born in a city in England.

- ⟨ Ada Lovelace, born in, English city⟩

**Alists.** We use alists as our main formal representation, however we expand the representation to add a natural language attribute written as *'nl'* followed be the phrase or the query that is being represented. We do this for better clarity of output when extracting alists from the models' outputs and matching them steps of reasoning. We also experiment with other formalisations as detailed below.

**Formal Logic.** We additionally experiment with the use of propositional and first-order logic to represent queries and reasoning.

*First-Order Logic Example:*

**Question:** What was the capital of Japan in 1960?

We define a predicate:

$Capital(x, y, z)$: "City $x$ was the capital of country $y$ in the year $z$"

and we can define our query as:

$\exists x\, Capital(x, \text{Japan}, 1960)$.

*Propositional Logic Example:*

**Reasoning:**
If I study hard, I will pass the exam.

- Let T be the proposition 'I study hard' and U be the proposition 'I pass the exam'.
- Premise 1: T → U.

If I pass the exam, I will celebrate.

- Let V be the proposition 'I celebrate'.
- Premise 2: U → V.

Therefore, if I study hard, I will celebrate.

- Conclusion: T → V.

## 3.1 MAINTAINING A STRUCTURED INTERNAL STATE

While previous work has altered the internal architecture of neural networks to incorporate symbolic methods (Riegel et al., 2020; Badreddine et al., 2022), we look at integrating structured reasoning into pre-trained LLMs through prompt fine-tuning and parameter-efficient fine-tuning (PEFT). Additionally, whereas related work on autoformalization with LLMs (Wu et al., 2022) addresses the challenge of translating natural language mathematics into formal specifications and proofs, our ap-

proach focuses on adopting structured representations in general reasoning tasks. We explore this problem using these two techniques:

**Prompt fine-tuning.** We prompt a reasoning model to utilise a given formalised representation at each step of its thinking. We use Chain-of-Thought (CoT) (Wei et al., 2022) prompting techniques to elicit step-by-step reasoning outputs from model. This provides added transparency and allows (depending on the representation itself) for the reasoning output to be verifiable. We experiment with zero-shot (with the exception of alists), one-shot and few-shot prompts.

**Parameter Efficient Fine-tuning (PEFT)** We use LoRA (Hu et al., 2021) to fine-tune a reasoning model on natural language and alist pairs. Since, to the best of our knowledge, there are no pre-existing datasets for our task, we generate our own (**ALIST8.5K**) using OpenAI's o4-mini (OpenAI, 2025a). We discuss the dataset generation in Section 4.3.

## 4 EXPERIMENTAL SETUP

This section details the experimental setup used to evaluate our proposed methods. We describe the baseline models, the benchmarks used for evaluation, the process for generating the synthetic alist dataset, and the hyperparameters used for fine-tuning. All code and supplementary materials[1] are available.

### 4.1 BASELINE MODELS

We experiment on the Qwen3-4B and Qwen3-8B models (Qwen, 2025) for our experiments, which are dense models that outperform state-of-the-art models of comparable size on a multitude of reasoning benchmarks. Both models utilize Grouped Query Attention (GQA)(Ainslie et al., 2023), SwiGLU activations (Shazeer, 2020), and Rotary Positional Embeddings (RoPE) (Su et al., 2023). The Qwen3 series shows significant improvement over its predecessors, for instance, on the GSM8K benchmark (Cobbe et al., 2021), Qwen3-4B-Base surpasses Qwen2.5-7B-Base (Yang et al., 2024). Similarly, the Qwen3-8B-Base outperforms the larger Qwen2.5-14B-Base on over half of the evaluated benchmarks. The instruction-tuned versions of these models benefit from a strong-to-weak distilliation pipeline which transfers knowledge from the larger Qwen3-32B and Qwen3-235B-A22B models. The models also feature a 'thinking mode' which allocates a larger token budget for complex, multi-step reasoning. In line with the report, all experiments are ran in thinking mode, with a sampling temperature of 0.6, a top-p value of 0.95, and a top-k value of 20.

In addition to these models, we experiment with the larger gpt-oss-20b model (OpenAI, 2025d), an autoregressive Mixture-of-Experts (MoE) transformer (Shazeer et al., 2017; Vaswani et al., 2023) with 20.9B parameters. The model features three different 'thinking modes' (low, medium, and high). In our experiments, we utilise the medium thinking mode. Due to its use of long CoTs, the model is particularly effective at mathematical reasoning. On the AIME 2025 benchmark, the model manages to match and even exceed the performance of the larger gpt-oss-120b model across the medium and high thinking modes respectively.

### 4.2 BENCHMARKS

We focus on two key benchmarks: GSM8K and MMLU. GSM8K is a dataset of 8.5K linguistically diverse maths word problems at grade-school level. We use this benchmark alongside MMLU (Hendrycks et al., 2021a;b), a massive multitask dataset of multiple-choice questions from various areas of knowledge across a range of difficulty levels, in order to provide a definitive assessment of our approach.

Our evaluation considers the impact of our methodology on core reasoning abilities. Consequently, we selected the aforementioned two benchmarks due to their diversity across topics and varying levels of difficulty. This deliberate focus led us to avoid state-of-the-art benchmarks like AIME or LongReason (Ling et al., 2025), which test for broader capabilities such as knowledge-base question

---

[1] https://osf.io/xmbz8/files/osfstorage?view_only=f15a51a5ff324b38920e335e81f84560

|  | GSM8K | MMLU |
|---|---|---|
| Qwen3-4B | 78.32 | 68.79 |
| Qwen3-8B | 76.42 | 75.01 |
| gpt-oss-20b | 84.99 | 84.23 |

Table 1: Accuracy results (%) of the baseline models on our chosen benchmarks.

answering and multilingual understanding, and where performance can be greatly limited by model size.

### 4.3 ALIST8.5K DATASET GENERATION

As mentioned previously, we use OpenAI GPT o4-mini to generate a synthetic dataset of natural language and alist pairs. However, due to the lack of diversity in the model's outputs in response to the generation prompt, using the model alone is not feasible when looking to create a varied, sizeable dataset for fine-tuning. Furthermore, while our few-shot prompt includes a range of examples, including mathematical reasoning, the model fails to generate any mathematical examples. Therefore, we use other datasets to supplement our natural language phrases and questions, and ask the LLM to generate their corresponding alists. Our key consideration when choosing these datasets is that we lack more complex natural language phrasing. Conversely, we lack any mathematical examples in our few-shot prompt, let alone complex ones. Therefore, we choose to use the LC-QuAd (Trivedi et al., 2017) dataset, due to the variety and the complexity of the questions in the dataset, and Deepmind's mathematics dataset (Saxton et al., 2019), which contains a variety of difficulty levels across a wide array of topics in mathematics. These provide natural language phrases/queries and we request the LLM to generate the corresponding alist representations. This supplies us with a total of 8.5k samples which we divide into train, validation and test splits.

### 4.4 FINE-TUNING SETUP AND HYPER-PARAMETERS

We experiment with LoRA rank and alpha values of 16 and 32. We train the model for 50 epochs, with a learning rate of 2e-5 and 2000 warmup steps.

## 5 RESULTS AND DISCUSSION

The baseline performance of Qwen3-4B, Qwen3-8B and gpt-oss-20b models on our two chosen benchmarks can be found in Table 1. In this section we discuss our results and their implications for integration of neuro-symbolic AI into pre-trained LLMs.

### 5.1 PROMPTING

We report the results of prompt-finetuning in Table 2 for first-order logic (FOL), propositional logic (Prop), semantic triples (SPO), and alists. We also provide an analysis of the *adherence score*, a measure of adherence to each requested structured representation in the models' outputs, in Table 3. The score is a simple percentage of the model outputs' which contain the prompted representation.

|  | FOL | | Prop | | SPO | | Alist | |
|---|---|---|---|---|---|---|---|---|
|  | GSM8K | MMLU | GSM8K | MMLU | GSM8K | MMLU | GSM8K | MMLU |
| Qwen3-4B | 77.78 | **67.22** | 77.18 | 65.62 | **80.67** | 66.43 | 76.27 | 67.15 |
| Qwen3-8B | 85.29 | **75.92** | 84.00 | 75.67 | 82.71 | 75.26 | **85.97** | 75.06 |
| gpt-oss-20b | **86.13** | 82.25 | 83.93 | 81.73 | 85.90 | **82.37** | 83.02 | 81.90 |

Table 2: Accuracy results (%) of prompting our chosen models to use a structured representation in their reasoning. Best accuracy across prompting strategies is reported for each formalisation. Best accuracies for each model and benchmark are in **bold**. Second best are underlined.

| | FOL | | Prop | | SPO | | Alist | |
|---|---|---|---|---|---|---|---|---|
| | GSM8K | MMLU | GSM8K | MMLU | GSM8K | MMLU | GSM8K | MMLU |
| Qwen3-4B | 33.81 | **33.18** | 7.51 | 11.89 | **55.88** | 78.74 | 56.56 | 30.57 |
| Qwen3-8B | 22.14 | **50.49** | 16.76 | 32.88 | 97.95 | 66.45 | **83.55** | 66.82 |
| gpt-oss-20b | **75.13** | 90.27 | 16.98 | **25.93** | 89.84 | 83.20 | 99.09 | 81.90 |

Table 3: Adherence (%) to constrained representation in the models' reasoning corresponding to experiments whose results are found in Table 2. Highest adherence scores for each formalism and each benchmark are underlined. Adherence scores corresponding to the highest accuracies are in **bold**.

For the smaller Qwen3-4B model, prompting for structured representation has a negligible impact on accuracy compared to the baseline. This is likely because the model struggles to consistently adopt the formalisms, as shown by its low adherence scores for First-Order Logic (FOL) and Propositional Logic (Prop) in Table 3. While adherence to the alist representation is relatively high (56.56%), inspection of the outputs show the model often fails to use it correctly, merely adopting the syntax without leveraging the structure for reasoning.

*Excerpt of Qwen3-4B Output:*

> Answer: 20*35*50 + 30*15*50 = 35000 + 22500 = 57500. Now, let's convert the solution into an alist. Alist:
>
> $$\{h : sum, v : \$x, s : 20, p : *, o : 35, t : 50, s : 30,$$
> $$p : *, o : 15, t : 50, s : \$x, p : +, o : \$x, t : year\}$$

The Qwen3-8B model shows a slight performance increase across all formalisms. However, this accuracy gain doesn't always correlate with strong adherence. For instance, adherence to FOL and Propositional Logic remains extremely low (22.14% and 16.76%, respectively), yet performance improves. Conversely, adherence is very high for SPO (97.95%) and alist (83.55%), suggesting the model can adopt these structures more easily, however it does not result in improved reasoning capabilities.

Across all models, we observe several key trends. Most significantly, Propositional Logic is consistently the least adhered-to structure, suggesting its syntax or application is the most challenging for the models to learn in-context. Furthermore, as model size increases, the general capacity to adopt and utilize an unfamiliar structure improves, which is evident in the higher overall adherence scores for gpt-oss-20b.

Use of the alist formalism is much higher across all models for the GSM8K benchmark. This is likely because the models misinterpret the syntax of formalism of alists as a simple variable substitution method, which aligns well with the step-by-step calculations often required to solve mathematical problems.

On the MMLU benchmark, across all models and representations we see that results remain stagnant. On the other hand, adherence to these representations differ greatly across formalisms and models. In fact, across both benchmarks, the best result (accuracy) almost never coincides with the highest adherence to a specific constrained representation. This suggests the models' reasoning is not affected by the use of these constrained representations, and furthermore that the models are not integrating these formalisms into their core reasoning processes.

## 5.2 FINE-TUNING

We report the results of finetuning Qwen3-4B to give Qwen3-4B-ALIST-16 and Qwen3-4B-ALIST-32 which are trained using rank and alpha values 16 and 32 respectively. Accuracy results are presented in Table 5 and adherence results can be found in Table 5. For the GSM8K benchmark, the fine-tuned Qwen3-4B-ALIST-16 model achieves the original model performance of 79.5% under the baseline condition, where it is not explicitly prompted to use the alist format. However, when

this same model is prompted to generate its reasoning in the alist structure, its accuracy drops significantly to 70.8%. The model's peak performance occurs when its adherence to the alist format is extremely low, at just 0.15%. Forcing adherence through prompting, which raises the score to 10.01%, hurts accuracy. We see a greater impact on the MMLU benchmark, where performance drops significantly. The fine-tuned models' peak performance is over 20% worse than the baseline model performance. Overall the fine-tuning results further show that adherence to a formalised structure do not correlate with performance. This further indicates that the fine-tuning is not sufficient in integrating the formalised structure into the model's reasoning process.

|  | GSM8K | | MMLU | |
|---|---|---|---|---|
|  | Baseline | Alist Prompt | Baseline | Alist Prompt |
| Qwen3-4B-ALIST-16 | **79.5** | 70.8 | 44.0 | 43.3 |
| Qwen3-4B-ALIST-32 | 78.5 | 57.5 | **46.4** | 45.6 |

Table 4: Results of fine-tuning Qwen3-4B on the alist representation with and without prompting the model to use alist. Best accuracies for each benchmark are in **bold**.

|  | GSM8K | | MMLU | |
|---|---|---|---|---|
|  | Baseline | Alist Prompt | Baseline | Alist Prompt |
| Qwen3-4B-ALIST-16 | **0.15** | 10.01 | 0.12 | 11.97 |
| Qwen3-4B-ALIST-32 | 0.08 | 17.66 | **0.10** | 13.49 |

Table 5: Adherence (%) to the alist representation in the finetuned models' reasoning corresponding to experiments whose accuracy results are found in Table 4. Adherence scores corresponding to highest accuracy are in **bold**. Highest adherence scores for each benchmark are underlined.

## 6 LIMITATIONS

Below we discuss the limitations of this project, and outline potential future research directions.

**Evaluation.** Our results and evaluation are restricted by our use of only two reasoning benchmarks. A direct next step for this project would be to explore a more extensive suite of state-of-the-art benchmarks, which include a greater array of complex reasoning tasks, for example, software engineering benchmarks (OpenAI, 2025c;b), different types of logic and question answering (Lin et al., 2025; Rein et al., 2023), and long context reasoning (Bai et al., 2024a;b; Ling et al., 2025).

**Verfying Model Outputs.** A key challenge we came up against in our research was the lack of verification of models' outputs. While it is possible to check the presence of syntax that indicates use of a formalism, it is non-trivial to verify that these outputs are correct representations of the accompanying natural language reasoning or the model's internal reasoning. Another key step would be to incorporate a verification step (e.g. using a external formal verification tool), which can lead to a more interpretable model.

**Dataset Generation.** Another limitation of our work is the dataset we've used for fine-tuning. The quality of our dataset is entirely dependent on the capabilities of our chosen model – in this case OpenAI's o4-mini. Due to resource limitations, this was the most feasible option, however further research should look at adding human verification into the dataset creation pipeline for better quality and to ensure dataset diversity. This could also mean extending the dataset to include examples from a larger range of reasoning tasks. Moreover, our dataset very simply trains the model on generating alist from natural language, however it does contain examples of where this representation is used consistently on reasoning. An import addition to the data would be step-by-step examples of using alists to reach an answer to a given question.

**Future work.** In this work, we only explore prompt fine-tuning and LoRA-based finetuning. However, full fine-tuning or architectural changes to integrate the symbolic and neural representations

into the model's reasoning, as suggested in (Nuamah, 2021), could result in an efficient method for improved reasoning and general question-answering abilities.

## 7 CONCLUSION

Our research investigates whether enforcing a structured internal state through formal representations can enhance the interpretability and trustworthiness of LLMs without compromising their reasoning performance. We explore two primary methods: prompt fine-tuning with various formalisms (first-order logic, propositional logic, semantic triples, and alists) and parameter-efficient fine-tuning using LoRA on a synthetic dataset of alist-natural language pairs.

Our findings demonstrate that constraining a pre-trained model like Qwen3-4B to these structured representations has a negligible impact on its core reasoning capabilities and can even be detrimental to performance. Across our experiments, we observed a significant disconnect between a model's adherence to a given formalism and its accuracy on reasoning tasks. While larger models showed a greater capacity to adopt the syntax of these formalisms, the best performance rarely coincided with the highest adherence scores, suggesting the models are not integrating these structures into their core reasoning processes.

This was particularly evident in our fine-tuning results, where forcing the specialized model to use the alist structure it was trained on caused a significant drop in accuracy compared to its baseline performance. Ultimately, our work highlights that simply compelling LLMs to produce structured outputs is insufficient for achieving neuro-symbolic reasoning.

Future research should explore more deeply integrated architectural changes to create models that are both highly capable in reasoning tasks but also transparent, producing verifiable outputs.

**Reproducibility Statement**  To ensure reproducibility of results in this work, we have share all code and relevant supplementary materials on OpenReview as part of the paper submission and via OSF at the following link: `https://osf.io/xmbz8/files/osfstorage?view_only=f15a51a5ff324b38920e335e81f84560`. The shared materials include a README file that clearly explains the data and code files needed to run the relevant experiments to reproduce our results.

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

## A    PROMPTS FOR FORMALISED REPRESENTATION

See Table 6 for examples of our prompting strategy on Qwen3-4B. The examples include prompts across three of the formalised representations: first-order logic, propositional, and semantic triples.

| Prompting Strategy | Example Prompt |
|---|---|
| Zero-shot | "Solve the following problem. Provide your reasoning both in natural language and utilising First-Order Logic at each step.Provide your reasoning before the final answer.Question: {question}" |
| One-shot | "Solve the following problem. Provide your reasoning both in natural language and utilising Subject, Predicate, Object triples at each step to form a knowledge graph. Provide your reasoning before the final answer.For example: Socrates is a human. SPO 1: Subject: Socrates, Predicate: is a, Object: human. All humans are mortal. SPO 2:Subject: Humans, Predicate: are, Object: mortal. Therefore, Socrates is mortal. SPO 3: Subject: Socrates, Predicate: is, Object: mortal. Question: {question}" |
| Few-shot | "Solve the following problem. Provide your reasoning both in natural language and utilising propositional logic at each step. Provide your reasoning before the final answer.Example 1: If it is raining, then the ground will be wet. Let P be the proposition 'It is raining' and let Q be the proposition 'The ground is wet'. Premise 1: $P \rightarrow Q$. It is raining. Premise 2: P. Therefore, the ground is wet. Conclusion: Q. Example 2: I will have either pizza or pasta for dinner. Let R be the proposition 'I will have pizza for dinner' and let S be the proposition 'I will have pasta for dinner'. $\neg$ R. Therefore, I will have pasta. Conclusion: S. Example 3: If I study hard, I will pass the exam. Let T be the proposition 'I study hard' and U be the proposition 'I pass the exam'. Premise 1: $T \rightarrow U$. If I pass the exam, I will celebrate. Let V be the proposition 'I celebrate'. Premise 2: $U \rightarrow V$. Therefore, if I study hard, I will celebrate. Conclusion: $T \rightarrow V$. Question: {question}" |

Table 6: Examples of prompting the model to use a factored representation in its reasoning

## B    EXAMPLE OUTPUTS OF FORMALISED REPRESENTATION

Please see Table 7 for examples of each formalised representation in model outputs.

## C    PROMPTS FOR SYNTHETIC DATASET GENERATION

Please see Table 8 for the prompts given to o4-mini to generate our synthetic alist dataset. The 'alist from natural language' prompt is used on the LC-QuAD and Deepmind mathematics dataset.

| Formalised Representation | Example Output |
| --- | --- |
| First-Order Logic | "Reasoning (First-Order Logic) Let

• $G$ be the government,
• $R(x)$ mean "$x$ recognizes natural rights,"
• $I(x)$ mean "$x$ commits an injustice."

Huemer's thesis can be formalised as: $\forall x, (\neg R(x) \rightarrow I(x))$ i.e. for any agent $x$, if $x$ fails to recognize natural rights, then $x$ commits an injustice. Option 3 states exactly this: "failures by the government to recognize natural rights count as injustices." Hence, the correct choice is 3." |
| Propositional Logic | "Let

• $P$: "The marked price is \$140."
• $Q$: "The discount rate is 5%."
• $R(x)$: "$x$ is the price after discount."

We also define a helper proposition:

• $S(y)$: "$y$ is the discounted price computed as price $\times$ (1 − discount)."

The logical steps are:

1. **Premise 1**
$$P \quad \text{(The price is 140).}$$
2. **Premise 2**
$$Q \quad \text{(The discount is 5\%).}$$
3. **Rule of Discount Calculation**
$$P \wedge Q \rightarrow S\big(140 \times (1 - 0.05)\big).$$
(If the price is 140 and the discount is 5%, then the discounted price is $140 \times 0.95$.)
4. **Apply the rule**
$$S\big(140 \times (1 - 0.05)\big) \rightarrow S(133).$$
5. **Conclusion**
$$R(133) \quad \text{(You will pay \$133)}".$$ |
| SPO | "Subject-Predicate-Object triples:

• Detector quantum efficiency – has – 0.1
• Photon detection – is – a Bernoulli trial
• Trial probability – equals – 0.1
• Number of photons – is – 100
• Total detections – follow – Binomial(100, 0.1)
• Binomial mean – equals – 10
• Binomial variance – equals – 9
• Standard deviation – equals – 3
• Correct answer – is – choice 3" |
| Alists | " ALIST description, Meaning

1. Desired purchase: {h:desired, v:?purchase, s:computers, n:500, p:price, o:\$700}, 500 computers at \$700 each.
2. Price increase, {h:increase, v:?newprice, p:price, o:\$700, f:percent, n:10}, 10% increase on \$700 $\rightarrow$ \$770.
3. Compute total, {h:calculate, v:?total, s:?purchase, p:quantity, o:500, f:times, o2:?newprice}, 500 $\times$ \$770 = \$385.000.

Answer: The company paid \$385,000 to buy the 500 computers at the new prices." |

Table 7: Examples of the gpt-oss-20b model outputs using a factored representation in its reasoning

| Prompting Strategy | Prompt |
|---|---|
| Alist from natural language | "An alist is defined recursively as a set of attribute-value pairs (x,y) to represent a question or fact such that x is either a functional, object-level or meta-level attribute and y is a constant, a variable, or an alist itself. Functional attributes are types of attributes that define the functional operation (arithmetic, statistcal or string) on an alist. Object-level attributes represent the object information that captures the meaning of a fact or question (such as subject, property, object, time and location). Meta-level attributes capture meta-information about the question or fact as well as meta-data that is generated when variables are instantiated via retrieval or inference. For example a simple alist of the question 'What was the capital of Japan in 1960?' would be {h:value, v:[?x], s:Japan, p:capital, o:?x, t:1960, nl:'capital of japan in 1960'} where s is a subject, p is a predicate and o is an object, t is time, ?x is a variable and h is the operation to perform and v is a list of variables and arguments for h. nl is the natural language meaning of the alist. Attributes v is not modified during reasoning. Another example: 'Country with the highest GDP in Europe?' would be {h:argmax, v:[?x,?y], s:?, p:GDP, o:?y, l:Europe, nl: 'Country with the highest GDP in Europe'}. Alists can be decomposed to create multiple new child alists whose instatiated variables are propagated to the parent alist. Each alist has an id attribute with a unique value and an parent_id attribute referencing its parent alist. Instantiated variables are also included in alists. Generate an alist from the following natural language: {phrase}.Provide your answer in the format: Alist: {alist}." |
| Alist and natural language pairs | "An alist is defined recursively as a set of attribute-value pairs (x,y) to represent a question or fact such that x is either a functional, object-level or meta-level attribute and y is a constant, a variable, or an alist itself. Functional attributes are types of attributes that define the functional operation (arithmetic, statistcal or string) on an alist. Object-level attributes represent the object information that captures the meaning of a fact or question (such as subject, property, object, time and location). Meta-level attributes capture meta-information about the question or fact as well as meta-data that is generated when variables are instantiated via retrieval or inference. For example a simple alist of the question 'What was the capital of Japan in 1960?' would be {h:value, v:[?x], s:Japan, p:capital, o:?x, t:1960, nl:'capital of japan in 1960'} where s is a subject, p is a predicate and o is an object, t is time, ?x is a variable and h is the operation to perform and v is a list of variables and arguments for h. nl is the natural language meaning of the alist. Attributes v is not modified during reasoning. Another example: 'Country with the highest GDP in Europe?' would be {h:argmax, v:[?x,?y], s:?, p:GDP, o:?y, l:Europe, nl: 'Country with the highest GDP in Europe'}. Alists can be decomposed to create multiple new child alists whose instatiated variables are propagated to the parent alist. Each alist has an id attribute with a unique value and an parent_id attribute referencing its parent alist. Instantiated variables are also included in alists. Below are some further examples: Example 1: The statement 'The genus of the primate whose hemoglobin protein exhibits the highest sequence similarity to that of Homo sapiens is Pan' can be represented by the alist h: AND, v:[fact_1, fact_2], fact_1:{h: value, p: genus, o: Pan, s: {h: argmax, p: hemoglobin sequence similarity, s: Homo sapiens, nl: The genus of the primate whose hemoglobin protein exhibits the highest sequence similarity to that of Homo sapiens is Pan}, fact_2:{p: genus, o: Pan, s: primate}}. Example 2: 'The comparison-based sorting algorithm with an average-case time complexity of O(n log n) and a worst-case space complexity of O(log n) is merge sort' can be represented by the alist {h: AND, v: [fact_1, fact_2, fact_3, fact_4], fact_1: {o: merge sort, p:type, s: sorting algorithm}, fact_2: {o: merge sort, p: type, s: comparison based}, fact_3: {h: average-case, p: time complexity, s: merge sort, o: O(n log n)}, fact_4: {h: worst-case, p: space complexity, s: merge sort, o: O(log n)}. Example 3. 'The integer closest to the value of Pi is 3' can be represented by the alist {h: round, s: Pi, o: 3}.Generate a natural language question or phrase and its corresponding alist. Provide your answer in json format structured as follows: Phrase: {phrase}. Alist: {alist}." |

Table 8: Prompts given to o4-mini for dataset generation

# D  EVALUATION RESULTS FOR DIFFERENT PROMPTING STRATEGIES

## D.1  QWEN3-4B

|  | FOL | | Prop | | SPO | | Alist | |
| --- | --- | --- | --- | --- | --- | --- | --- | --- |
|  | GSM8K | MMLU | GSM8K | MMLU | GSM8K | MMLU | GSM8K | MMLU |
| Zero-shot | 77.71% | **67.22%** | 77.18% | 65.59% | 80.67% | 64.43% | - | - |
| One-shot | 77.63% | 66.49% | 76.12% | 65.62% | 78.70% | 66.43% | 76.27% | 67.15% |
| Few-shot | 77.79% | 65.10% | 75.51% | 65.55% | 77.86% | 64.42% | 75.06% | 66.50% |

Table 9: Results of prompting Qwen3-4B to use a structured representation in its reasoning. Best accuracies for each benchmark are in **bold** and for each prompting strategy are underlined.

## D.2  QWEN3-8B

|  | FOL | | Prop | | SPO | | Alist | |
| --- | --- | --- | --- | --- | --- | --- | --- | --- |
|  | GSM8K | MMLU | GSM8K | MMLU | GSM8K | MMLU | GSM8K | MMLU |
| Zero-shot | 85.29% | **75.92%** | 83.70% | 74.56% | 82.56% | 75.26% | - | - |
| One-shot | 84.53% | 73.76% | 84.00% | 75.67% | 82.71% | 73.01% | **85.97%** | 75.06% |
| Few-shot | 83.40% | 69.87% | 83.32% | 71.41% | 81.73% | 71.99% | 84.08% | 74.88% |

Table 10: Results of prompting Qwen3-8B to use a structured representation in its reasoning. Best accuracies for each benchmark are in **bold** and for each prompting strategy are underlined.

## D.3  GPT-OSS-20B

|  | FOL | | Prop | | SPO | | Alist | |
| --- | --- | --- | --- | --- | --- | --- | --- | --- |
|  | GSM8K | MMLU | GSM8K | MMLU | GSM8K | MMLU | GSM8K | MMLU |
| Zero-shot | **86.13%** | 82.25% | 83.93% | 81.73% | 85.90% | **82.37%** | - | - |
| One-shot | 83.09% | 81.45% | 83.09% | 81.13% | 82.94% | 82.07% | 83.02% | 81.90% |
| Few-shot | 83.85% | 81.26% | 82.79% | 81.01% | 83.55% | 81.21% | 83.02% | 81.66% |

Table 11: Results of prompting gpt-oss-20b to use a structured representation in its reasoning. Best accuracies for each benchmark are in **bold** and for each prompting strategy are underlined.

## E    FINE-TUNING QWEN3-4B

In Figures 1 and 2 we present the plots for training and validation loss for both fine-tuned models.

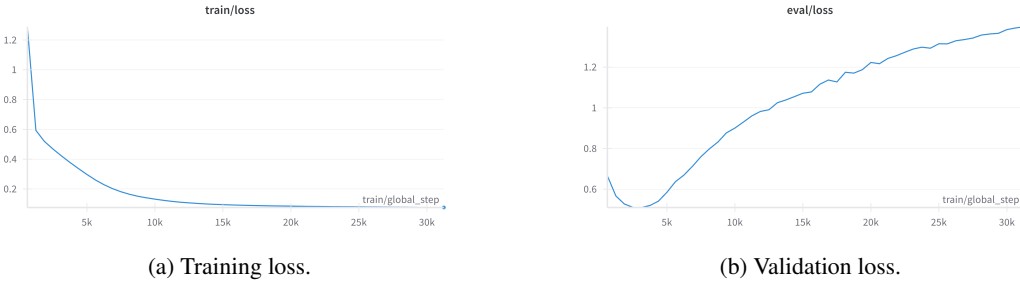

(a) Training loss.                                      (b) Validation loss.

Figure 1: Training and validation loss for fine-tuning with LoRA rank and alpha values 16.

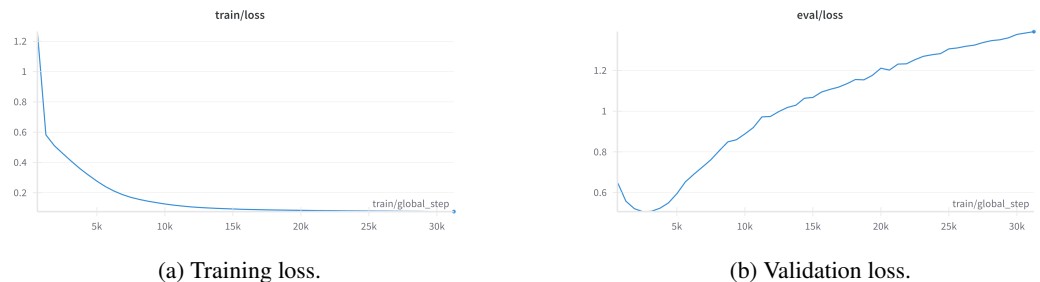

(a) Training loss.                                      (b) Validation loss.

Figure 2: Training and validation loss for fine-tuning with LoRA rank and alpha values 32.