# OpenReview forum: "Factored Representation for Neuro-Symbolic AI"
_ICLR.cc/2026/Conference — Submitted to ICLR 2026_

### Official Review · Reviewer_7ZQH · 2025-10-28

**Soundness:** 2
**Presentation:** 2
**Contribution:** 2
**Rating:** 4
**Confidence:** 3

**Summary:**

The paper provides an empirical exploration of neuro-symbolic integration in LLMs, employing diverse formalisms and fine-tuning strategies across multiple model sizes. The findings reveal persistent challenges in model adherence to formal structures, limiting interpretability gains despite maintained performance. While the work provides valuable insights into the separation of generative and reasoning capabilities in LLMs, the reliance on a narrow set of benchmarks and synthetic data generation raises questions about broader applicability.

**Strengths:**

1.	The introduction frames the opacity of LLM decision-making as a trustworthiness barrier in complex reasoning, linking it to neuro-symbolic AI's potential for explicit logic integration.
2.	All code, datasets, and hyperparameters are openly shared, with detailed descriptions of fine-tuning setups and dataset generation.

**Weaknesses:**

1.	The paper includes numerous detailed examples for formalisms (e.g., alist Examples 1–3, semantic triple example, FOL and propositional logic examples), which collectively span significant space in Sections 2.2 and 3, detracting from core methodological exposition.
2.	Evaluation relies solely on GSM8K and MMLU, omitting more complex reasoning benchmarks (e.g., MATH, GPQA, TheoremQA and BBH), which risks underrepresenting the formalisms’ utility in broader scenarios.
3.	The experiments primarily evaluate Qwen3, without testing on alternative architectures like Llama or other dense transformers, which limits claims about generalizability of structured representations across diverse LLM backbones.
4.	Adherence score is defined as percentage of outputs containing the prompted representation without parsing for semantic correctness, reducing experimental rigor.
5.	While adherence scores measure syntactic compliance, no method is provided to verify whether structured outputs semantically align with reasoning steps or are correct.
6.	Fine-tuning results in Table 4 show performance drops when enforcing alists, but the paper does not explore why adherence increases yet accuracy decreases.

**Questions:**

Please refer to the weaknesses.

---

### Official Review · Reviewer_rt5d · 2025-11-01

**Soundness:** 2
**Presentation:** 2
**Contribution:** 2
**Rating:** 4
**Confidence:** 3

**Summary:**

Investigate whether enforcing a factored representation during finetuning through prompt-tuning and lora tuning affects the LLM's reasoning ability and whether it can enhance interpretability. It found that (1) state-of-the-art models struggle to generate consistently structured reasoning, (2) their core reasoning capabilities remain largely intact.

**Strengths:**

1. combining factored representation in tuning to make LLMs output formal reasoning is a promising way to get interpretable

**Weaknesses:**

**1. First, I am not an expert in this domain.**

2. I don't think the findings got by this paper is very meaningful for further model development. The model struggles to generate consistently structured reasoning, but why? The author should tell readers if this is because **(1) our training method is not good enough or (2) training methods to combine Neuro-Symbolic with LLMs just don't work completely and we need to find something new  (we maybe should propose some new NN architecture?)** as an analysis paper, and **what should we do to make formal reasoning successful for LLMs, according to your experiments**.

for (1), one questionable experimental setting is that in this paper, **only prompting (5.1), prompt tuning & LoRA tuning are used (tuning used only in Qwen3-4B), but all these methods are too weak to make LLMs learning, Qwen3-4B is also too small to get meaningful results**, I believe **full tuning should be implemented on larger models** to verify if today's LLM could not leverage formal reasoning. Otherwise, claims like "show that the models largely struggle to adhere to the provided structure" in the introduction should be presented.


3. writing: Table 1 & Table 2 should be merged to be compared easily for readers

After double checking the paper, I improve confidence score to 3.

**Questions:**

see weakness

---

### Official Review · Reviewer_P31K · 2025-11-02

**Soundness:** 1
**Presentation:** 2
**Contribution:** 2
**Rating:** 0
**Confidence:** 4

**Summary:**

**Summary**

The authors attempt to encourage rule-based reasoning in pre-trained LLMs through prompt fine-tuning and LoRA SFT. Their motivation is the insight that constraining the internal representations of the model during CoT should improve performance on tasks like mathematics which should theoretically benefit from logically structured CoT traces. Additionally, they argue that such an approach should lead to more interpretable and verifiable outputs from the model.

**Initial Recommendation: Reject**

I do not believe that the authors’ approach addresses their proposed hypothesis, nor do I believe that their results rise to the level of supporting the conclusions that they make. A large part of the authors’ work is in prompt fine-tuning, which has been explored in similar contexts. I do not believe that the current “story” of the paper is convincingly novel. Finally, many key details about the dataset, training, methodology, and evaluation are missing which makes drawing conclusions about the authors’ results challenging.

**Supporting Arguments**

The authors’ primary hypothesis is “forcing a pre-trained LLM to maintain a structured internal state using a formal representation can lead to comparable reasoning performance, but with significantly improved interpretability”.

Their results do not support their hypothesis, neither do they disprove the hypothesis. Their prompted models underperform or barely overperform the baseline, and these prompted models do not satisfactorily follow the desired neuro-symbolic structure. Their LoRA SFT models universally underperform the baseline models and struggle to follow the desired CoT structure. Despite this, they draw vague and unsubstantiated conclusions like “Propositional Logic is consistently the least adhered-to structure, suggesting its syntax or application is the most challenging for the models to learn in-context” (line 357). This conclusion ignores the obvious null hypothesis that their prompting for propositional logic was poorly designed, and the authors make no attempts to address this. Another unsubstantiated claim is that “simply compelling LLMs to produce structured outputs is insufficient for achieving neuro-symbolic reasoning” (line 452). This claim is unsubstantiated by the work, again because of the null hypothesis. Furthermore, my impression is that this claim is not supported by the literature, where I have found examples of works which successfully implement methods which compel the LLM to produce structured outputs (see below). To substantiate their claims the authors would need to show that outputs in neuro-symbolic reasoning are somehow different from outputs in other structured formats, like JSON, mathematical logic, or code, where previous authors have shown positive results.

Previous works have attempted similar prompt fine-tuning strategies with successful results. For example, “Structured Chain of Thought” by Li et al. 2025 uses prompting to steer the CoT reasoning chain to adhere to a specific output structure, in this case coding logic. “Faithful Chain of Thought” by Qing et al. 2023 actually “forces” the structure of the internal representations and is verifiable. Next, “Let’s Verify Step by Step” by Lightman et al. 2023 uses supervised training to steer a model towards having verifiable CoT steps. Without doing too much of a detailed literature review for the authors, I will also mention “Learning to Generate Structured Output with Schema Reinforcement Learning” by Lu et al. 2025 and "Program of Thoughts Prompting” by Chen et al. 2022. All of these works are conspicuously absent from the paper’s literature review and references. I think that while none of these previous studies do the exact same thing, many are similar enough to warrant an explanation from the authors as to why their approach is substantially different, and more importantly, why their negative results should be considered stronger than previous authors who show positive results.

**Additional Feedback**

I think that this project is a promising start but does not yet rise to the level of being accepted to a conference. In order to make a solid contribution to the literature in this area I recommend the authors do the following:

1. Further explore the literature and find previous authors who have done projects forcing specific representational structure in CoT. Use these as a baseline for your study.
2. Re-frame the project as “We explore the efficacy of using prompt fine-tuning and LoRA SFT to encourage pre-trained LLMs to have CoT traces which adhere to a specific neuro-symbolic structure”. Avoid making strong conclusions from the model performance in cases where there is a null hypothesis.
3. If the models are performing poorly, explore hypotheses which are testable and differentiate between material properties of the LLM CoT process and potential implementation details.
4. Explore the literature regarding LoRA fine-tuning more deeply. I am pretty sure that you will be able to get a LoRA fine-tuned model to have high adherence and high accuracy, but it may require more involved training and / or better data. Getting the LoRA fine-tuned models to beat the baseline models would represent a novel contribution to the literature, in my opinion. Also this would be quite interesting if you can get it working.

Other thoughts:

- You suggest that the prompt fine-tuning and LoRA SFT are “forcing” the model to “maintain a structured internal state” (line 151). I don’t think it is fair to call this “forcing”. I think a better word might be “encourage”. There are methods in the literature for explicitly forcing the model to adhere to specific formats using verifiability techniques, and this feels like a more appropriate use of the word. I was confused upon my first reading by the use of this word.
- I haven’t touched on the dataset in this review, but I think that the dataset could be improved too. I am very concerned by the use of a non-symbolic model to attempt to generate symbolic training data in an unsupervised and unverifiable context. I think that this needs to be addressed and a compelling argument presented as to why the data is sufficient for this task.

Minor typography issues (did not affect score):

- Line 37: Uses “AI” but defines acronym “AI” on line 70.
- Line 263: Remove “in order to provide a definitive assessment of our approach”.
- Line 376: First instance of “Table 5” should be “Table 4”.
- Lines 389-390: Best model (baseline) is not bolded.
- Lines 398-399: Best model (baseline) is not bolded.
- Line 528: Extra space between the “3” and the “rd” in “3rd”.

**Strengths:**

- Hypothesis is reasoned, motivated, and grounded.
- LoRA fine-tuning as a way to encourage a model to have structured CoT appears novel.

**Weaknesses:**

- The central hypothesis is not substantiated (see above).
- The key metric of the paper, “adherence”, is not defined.
- The work is not particularly novel. There are many works on “chain of thought structured reasoning” which (a) were not cited, (b) do similar or the same thing, and (c) have substantially better results (see above).
- Because there are other works which show better results than this paper, the methodological approach is called into question and it is unclear if the results found by the authors are poor because of a fundamental limitation with the LLM (in which case the authors should explore this further) or poor due to the authors’ implementation (null hypothesis).
- Performance is only measured on two benchmarks: GSM8K and MMLU. They justify not including more benchmarks as follows: “This deliberate focus led us to avoid state-of-the art benchmarks like AIME or LongReason (Ling et al., 2025), which test for broader capabilities such as knowledge-base question answering and multilingual understanding, and where performance can be greatly limited by model size”. However, because they are comparing SFT and prompting performance, it should not matter what the benchmark content is. The authors’ hypothesis can be tested on any model of any size as long as we compare with a fair baseline.
- The authors do not measure “interpretability” of the model’s outputs or CoT, nor do they even propose a method or metric for doing so. This important analysis is crucial to testing the authors’ central hypothesis (“significantly improved interpretability”).
- Tables do not include the baseline results. The baseline results are especially important for comparing performance and adherence and not including the baselines leads to misleading results. For example, Table 2 indicates that the Qwen3-4B trained with FOL is the best performer on MMLU. However this model underperforms the baseline, which is absent from the table.

**Questions:**

- How is adherence measured?
- What are results on other reasoning benchmarks?
- How was the dataset generated? Section 4.3 does not sufficiently address this.
- How was the model **trained**!? Section 4.4 discusses a few hyperparameters, but notably
    - There is no mention of the training target for the fine-tuning.
    - There is no mention of how they tried to enforce CoT adherence during SFT. The limitation section leads me to believe that they **did not** enforce CoT adherence during SFT.
- Why didn’t you early stop at the lowest eval loss for the fine-tuning? It looks like the best performance was around 3k steps, but the model is wildly over-trained to 30k steps. What does performance look like at the 3k checkpoint? (see training curves in appendix).
- How does your method compare to other methods which use prompt fine-tuning and SFT / RL to encourage adherence to structured outputs? Is there something different about neuro-symbolic structure which is more challenging for LLMs than code or JSON?

---

### Official Review · Reviewer_kPcn · 2025-11-04

**Soundness:** 2
**Presentation:** 2
**Contribution:** 1
**Rating:** 2
**Confidence:** 3

**Summary:**

This paper forces LLMs to encode intermediate states in symbolic forms (triples, attribute–value lists, first-order logic) to boost interpretability without hurting reasoning; although models struggle to generate consistently well-structured traces, core reasoning remains intact—implying generation and reasoning aren’t fully aligned and motivating specialized models with verifiable chains of thought.

**Strengths:**

1. The paper writing is largely clear.

**Weaknesses:**

1. I find it difficult to discern the practical significance of this paper. Put differently, the conclusions are not surprising: the natural semantic space is vastly larger than the space of formal languages, and many real-world problems cannot be reduced to formal language in the first place. Consequently, no matter how much abstraction is applied, under the current setup the work feels rather “toy,” leaving the study lacking in both significance and scalability.

2. The experimental design fails to demonstrate genuine interpretability. Even with the output format constrained, it remains explicit—you cannot infer the model’s internal state regularities, nor is it clear whether the model truly grasps formal abstraction or is merely mimicking. Moreover, the formalization approaches discussed in the paper—such as first-order logic—are themselves quite simple in form. As a result, it is difficult to draw conclusions of substantive value.

3. The paper’s approach to abstracting natural semantics into formal language is very similar to [1] and should be explicitly discussed.

4. The work lacks evaluation of modern reasoning models—for example, DeepSeek-R1.

[1] Meta-Reasoning: Semantics-Symbol Deconstruction for Large Language Models.

**Questions:**

N/A

---

### Meta-Review · Area_Chair_8QB7 · 2025-12-24

**Summary:**

This paper presents an exploration study about the impact of formalizing the LLM’s internal state structure on LLMs’ decision-making performance. It leverages prompt fine-tuning and LoRA with various structured formalisms. The experiment results demonstrate that constraining models to a formal representation does not affect the reasoning capabilities.

All the reviewers have negative scores. They raised several major concerns, as summarized below.

1.	The significance of this paper is restricted. The findings are not surprising, and not helpful for realistic scenarios.
2.	The methods employed and experiment results cannot fully support the authors’ hypothesis and arguments
3.	Some relevant works are missing, as given by the reviewers.
4.	The evaluation benchmarks are limited. Some metrics have unclear definitions.

Those raised points are critical and hard to address. The authors did not provide the responses to them. Based on those comments, AC recommended rejection.

**Reviewer Concerns:**

The authors did not provide a rebuttal

**Reviewer Scores:**

The authors did not provide a rebuttal

---

### Decision · Program_Chairs · 2026-01-26

Reject